# Research

analytical chemistry/chemical engineering/energy

particle size distribution, seepage experiment, migration resistance, profile control

**Author for correspondence:**
Bin Huang
e-mail: huangbin502a@163.com

This article has been edited by the Royal Society of Chemistry, including the commissioning, peer review process and editorial aspects up to the point of acceptance.

# The efficiency of migration and profile control with emulsion systems in class III reservoirs

Cheng Fu[1,2,3], Tingting Zhu[1], Bin Huang[1,2], Tingge Dai[1], Ying Wang[4], Wei Zhang[5] and Xiangbin Liu[6]

[1]Northeast Petroleum University, Daqing, Heilongjiang, People's Republic of China
[2]Unconventional Oil and Gas Science and Technology Research Institute, China University of Petroleum, Beijing, People's Republic of China
[3]Post-Doctoral Scientific Research Station, Daqing Oilfield Company, Daqing, People's Republic of China
[4]Aramco Asia, Beijing, People's Republic of China
[5]China University of Petroleum Huadong, Dongying, Shandong, People's Republic of China
[6]Research Institute of Production Engineering, Daqing Oilfield of CNPC, Daqing, People's Republic of China

CF, 0000-0003-3335-2873; TZ, 0000-0003-3883-2657;
BH, 0000-0001-6158-8678; TD, 0000-0003-4117-8537;
YW, 0000-0002-8954-0796

In order to reveal the migration law and profile control efficiency of the emulsion system in the core of class III reservoir (the permeability less than 100 mD), the influence of the size and distribution of the droplets, the effect of the migration law and the adjustment of the interlayer contradiction to class III reservoir are studied in this paper. By measuring the droplet distribution in the emulsion system, we found that the average droplet diameter decreases with the increase in water cut. But, the droplet distribution is the most uniform when the water cut is 50%, which is the transition point of the emulsion. Through the core seepage test, the pressure is measured when the emulsion system is flowing through the core. It can be seen that the emulsion flowing is related to the resistance coefficient, the viscosity of the chemical agent and the core permeability; that is, the greater the resistance coefficient, the greater the viscosity of the chemical agent, the smaller the core permeability is and the higher the level of the emulsion is. The matching chart between different emulsion systems and layers with different permeability of class III reservoir was established. The effect of profile control with different permeability contrasts was predicted according to the matching chart. The emulsification has a positive effect on the enhancement of recovery when the permeability contrast is small, but a negative effect when the

permeability contrast is larger. The study of the emulsion migration and profile control efficiency in class III reservoir are of great significance for understanding the emulsification in class III reservoirs.

## 1. Introduction

Flowing of emulsion in porous media is an important phenomenon in oil production, which is also an interesting topic in many applications in science and engineering, especially the petroleum industries. According to the statistics, 80% of the crude oil in the world is produced in the form of emulsion [1,2]. There are many operations related to the emulsion during the development of the oil field, such as drilling fluid in the drilling process, completion fluid in the completion process, migration of crude oil in the formation, emulsion formed through the interaction of chemical system and crude oil, crude oil transported in a pipe, etc. Currently, with deeper and deeper understanding of emulsion in enhanced-oil recovery (EOR), the recovery rate has been improved during the tertiary oil recovery process. Therefore, it is significant to study the seepage law and the migration of emulsion in formation [1,3]. The proportion of emulsion system and its related technologies is increasing in oil stabilization and production control in high water cut oilfield. With increasing water cut in the reservoir and the complexity of water flooding, there are increasing technical requirements for this field. The development of the emulsion system and its related technologies are also promoted [4,5]. In recent years, many new developments have been made in the research and application of emulsifiers, including emulsion flooding, spontaneous emulsification flooding, micro emulsion flooding, emulsion for profile control, emulsified heavy oil for water control and other related technologies. These are important in improving water-flooding efficiency and increasing recovery for high water cut oil fields in China [6,7]. McAuliffe conducted laboratory studies by injecting an emulsion made with crude oil and diluted sodium hydroxide into sandstone cores. Reduced water permeability was observed even after many pore volumes of water injection after the emulsion injection. Then a field test was reported that had positive responses of flood pattern and incremental oil recovery by emulsion flooding, which were caused by plugging the oil droplets at high permeability zones and improving reservoir heterogeneity. McAuliffe is a pioneer in the area of emulsion research. The parameters affecting crude oil emulsion stability were studied by Sun *et al.*, and include rate of revolution, emulsifying time, water cut, water conductivity and the temperature at which the water-in-oil emulsion was prepared in laboratory. The results show that the rate of revolution is a key factor for generating crude oil emulsion [8]. Experiments were performed to characterize the emulsions in terms of their physiochemical properties and size distribution of the dispersed oil droplet in water phase by Mandal to investigate the efficiency of oil–water emulsions in EOR. Substantial additional recoveries (more than 20% of original oil in place) over conventional water flooding were achieved in the present investigation [9]. The ratio of displaced oil and the produced fluid in different layers was studied on triple parallel heterogeneous cores with different permeability to simulate the underground emulsification process. The result showed that greater permeability variation coefficient caused greater the oil recovery and reduction in the produced fluid in high permeability layers [10]. However, very few studies were found on the properties of emulsion formed in the process of oil displacement under reservoir conditions. Most previous research mainly focused on the influence factors, such as formation water, crude oil and surfactant on the stability of emulsion, etc. [11–13]. The purpose of most of the research is to solve the problems of processing and gathering of liquid production. The research on characterizing migration of the emulsion formed in the porous media is not completed and further research is needed [14–16]. Therefore, characterizing migration properties of the emulsion in layers are studied in this paper. The flow of complex fluids through porous media results in multiphase flow displacement at different scales, from pore-level to Darcy scale. Experiments have shown that injection of oil-in-water emulsions can be used as an effective EOR method, leading to substantial increase in the volume of oil recovered [17–19]. From the literature, pore-scale flow visualization as well as core flooding results have demonstrated that the enhanced recovery factor is regulated by the capillary number of the flow [20,21]. However, limited research has been done on the effect of emulsion on the profile control in heterogeneous reservoirs. Therefore, the profile control of emulsion in different layers is studied in this paper.

This paper mainly introduces the stability of emulsion, the migration characteristics of emulsion in porous media and the efficiency of emulsion on EOR. By means of theoretical analysis and laboratory experiment, the change of particle size at different water saturation is measured by microscope and

**Table 1.** Formula of the prepared brine.

| salinity (mg l$^{-1}$) | sodium chloride (g l$^{-1}$) | potassium chloride (g l$^{-1}$) | calcium chloride (g l$^{-1}$) | magnesium sulfate (g l$^{-1}$) | sodium sulfate (g l$^{-1}$) | sodium bicarbonate (g l$^{-1}$) |
|---|---|---|---|---|---|---|
| 6778 | 3.489 | 0.020 | 0.064 | 0.262 | 0.114 | 2.829 |

particle size analyser. Starting with the migration of emulsion in the reservoir pores, the microscopic percolation of emulsion in different layers is studied. Combined with the core percolation experiment, the resistance of the emulsion in the migration process is considered and the macroscopic percolation characteristics of the emulsion in the oil layer are studied. The effect of emulsion flooding on EOR is predicted by the matching chart established with the relationship between different emulsion systems and layers with different permeability in class III reservoirs. The results of this paper are helpful to explain the percolation mechanism and the effect of profile control of the emulsion in porous media of the class III reservoirs in essence. This research has great value for any theoretical study in this field.

# 2. Material and methods

## 2.1. Materials

Oil: kerosene provided by the No. 1 Daqing Oil Production Plant.

Brine: prepared by adding sodium chloride, potassium chloride, calcium chloride, magnesium sulfate, sodium sulfate and sodium bicarbonate into distilled water. The salinity is 6778 mg l$^{-1}$. The specific formula is shown in table 1.

Core: cylindrical homogeneous cores with a diameter of 25 mm and a length of 100 mm. The effective permeability is 20, 40, 60 and 80 mD.

Emulsifier: Span80 with the concentration of 1%, 1.5% and 2%.

## 2.2. Methods

### 2.2.1. Emulsion preparation

Operation schemes:

(1) At different water cut, 1% Span80 was mixed with kerosene. Then the mixture was stirred by ZLE-B500 homogenizer (Shanghai Zhongshi Machinery Limited Company). The resistance of the emulsion with different water cut, 20%, 30%, 40%, 50%, 60% and 70%, was compared.
(2) Span80 of different concentrations, 1, 1.5 and 2%, was mixed with kerosene at water cut of 60%. Then the mixtures were stirred by the homogenizer. The resistance of the emulsion with different surfactant concentrations was compared.

Operation steps:

(1) Add the configured water phase and kerosene to the beaker.
(2) Put the beaker into the water bath at 45°C for 15 min.
(3) Stir for 5 min (11 000 r.p.m.) by the homogenizer and place the emulsion in the oven at 45°C.
(4) Repeat the above steps after changing the oil–water ratio or the surfactant concentration.

### 2.2.2. Measurement of particle size distribution

The prepared emulsion system was loaded into the measuring cylinder of the MS2000 laser particle size analyser (LPSA) (British Malvern Instruments Limited Company) to test the particle size distribution. Operation steps:

(1) Turn on the switch of the apparatus and start the software in the computer after 15–20 min of preheating.

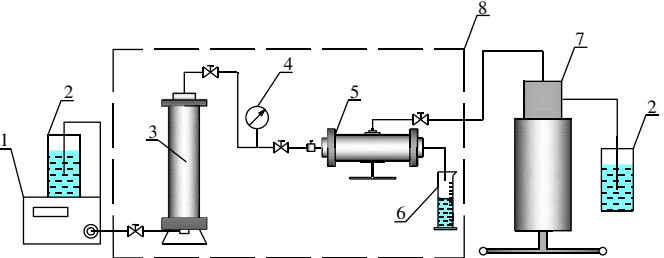

**Figure 1.** Experimental-process diagram. 1, plunger pump; 2, beaker; 3, accumulator; 4, pressure gauge; 5, core holder; 6, graduated cylinder; 7, manual pump; 8, oven.

**Table 2.** Composition of different emulsion systems.

| water cut (%) | 20 | 30 | 40 | 50 | 60 | 70 |
|---|---|---|---|---|---|---|
| surfactant concentration (%) | 1 | 1.5 | 2 | | | |
| core permeability (mD) | 20 | 40 | 60 | 80 | | |

(2) Turn on the pump and the ultrasonic vibrator to check whether the equipment is running normally.
(3) Set different pump speeds based on the different properties of the samples.
(4) Determine whether to turn on the ultrasonic instrument and what strength to use according to the properties of the samples.
(5) Set the optical parameters, number of the test sample. Then use the secondary water to measure the background of the sample.
(6) Add dispersed sample to control the concentration in the test range after background measurement. Begin measurement when the dispersed system is stable.
(7) Collect data and process necessary data.
(8) Remove the solution in the pipe and the sample slot completely after the LPSA tests finish. Clean the sample slot and the pipe with the secondary water for the next measurement.
(9) Turn off the power and soak the blender with secondary water after the test.

### 2.2.3. Core seepage experiment

Different preparation conditions will cause different distribution of the droplets in the dispersive phase of the emulsion system, which cause different rheological properties of the emulsion and produce different migration conditions in the core. The migration of emulsion systems with different water cut in class III reservoirs was studied.

The composition of different emulsion systems is shown in table 2. The migration of emulsion systems with different water cut in class III reservoirs was studied. The experimental schema is shown in figure 1.

Core seepage experiment with a chemical agent prepared by emulsion system:

(1) Record the dry weight of cylindrical cores.
(2) Vacuum these cores for about 4 h.
(3) Saturate cores with the simulated formation water with salinity of 6778 mg $l^{-1}$, which is similar to that of the Daqing oilfield. Record the core quality after water saturation. Calculate the pore volume of the core.
(4) Measure the core permeability with water. The entrance of the core holder is attached to the pressure meter which is set to zero. Three displacements are set through 2PB00C plunger pump (Beijing Satellite Manufacturing Plant). The entrance pressure of the core holder is recorded after the pressure is stable. The outlet pressure is the atmospheric pressure. According to Darcy's Law, the permeability and the average value of three permeabilities can be calculated.
(5) Inject emulsion. At 45°C, the emulsion is injected at 0.3 ml min$^{-1}$ until the pressure is stable. The pressure is recorded for every 0.5 pore volumes of injected emulsion.
(6) Inject water for the same pore volumes as to the emulsion. Record injection pressure for every 0.5 pore volume of injected water.

The change of transport resistance of emulsion system in porous media is obtained and resistance curve of emulsion migration is plotted. Resistance is characterized by resistance coefficient $F_R$

$$F_R = \frac{k_w \times \mu_e}{k_e \times \mu_w} = \frac{\Delta p_e}{\Delta p_w}.$$

where $\Delta p_e$ is the pressure during emulsion injection, MPa; $\Delta p_w$ is the pressure in the post-water injection, MPa.

# 3. Results and discussion

According to the particle size distribution and the results from the core seepage experiment, the formation and effect of the emulsion system in the layer can be obtained.

## 3.1. Existence of droplets

As dispersed fluid, the distribution of the emulsion droplets in the pores can be roughly divided into three types:

(1) When the droplet diameter is much smaller than the pore throat, it passes directly. It can be regarded as a continuous phase (figure 2a). In this situation, the particle size of the emulsion droplets in the porous media remains almost unchanged.
(2) When the droplet diameter is similar to the pore throat (figure 2b), the emulsion droplet is blocked in the porous medium. The average particle size decreases slightly. The plugging–plug removal–plugging phenomenon is obvious.
(3) The emulsion droplets are cut very small due to demulsification or fragmentation when the emulsion droplet passes through the smaller pore throat (figure 2c). The emulsion is the mixture of oil and water when migrate in the porous medium. The resistance is small due to small scattered droplets.

## 3.2. Particle size distributions of different emulsion systems

The initial and the outlet (after flow through the cores) particle sizes of the emulsion for different water cut and surfactant concentrations are shown in tables 3 and 4. When the emulsion flows through the cores with permeability of 20 and 40 mD, there are three conditions for the ratio of the droplet diameter to the pore diameter: De/Dp < 1, De/Dp ≈ 1 and De/Dp > 1. However, when the emulsion flows through the cores with permeability of 60 and 80 mD, there is only one situation: De/Dp < 1. The droplet diameter increases at first and then decreases with increasing water cut. When the emulsion flows through the cores with permeability of 20 mD, there are two conditions for the ratio of droplet diameter to the pore diameter: De/Dp > 1 and De/Dp ≈ 1. When the emulsion flows through the cores with permeability of 40 mD, there are also two conditions: De/Dp ≈ 1 and De/Dp < 1. However, when the emulsion flows through the cores with permeability of 60 and 80 mD, there is only one condition: De/Dp < 1. The droplet diameter increases steadily with the surfactant concentration. (De refers to the droplet diameter of the emulsion; Dp refers to the pore diameter.)

The difference of outlet and the original droplet size is very small when De/Dp < 1 and De/Dp ≈ 1. It shows that the average pore diameter of the porous medium is greater than the particle size of the emulsion droplet. The emulsion droplet will not be stuck in the pore throat and can migrate freely in the pore. The droplets migrate freely in larger pores due to smaller resistance. The emulsion can pass through a relatively large pore by deformation. However, due to adsorption on the pore surface, the migration resistance of emulsion increases with the increase in emulsion injection volume. The migration of emulsion in porous medium at this time is basically the migration of oil and water and dispersed small liquid drops. Therefore, the migration resistance of the emulsion in the cores gradually increases at first and then becomes stable when the emulsion breaks through.

The difference of the outlet and the original droplet diameter is great when De/Dp > 1. The larger the original droplet is, the more the emulsion droplet decreases. The reason is that larger droplets initially block smaller pores resulting in pressure increase. The large droplets pass through pore throat by deforming. Larger droplets break into smaller droplets or demulsification when the injection pressure accumulates to a certain level. As a result, the size of the outlet droplet decreases with the increase in the size of the injected emulsion droplet.

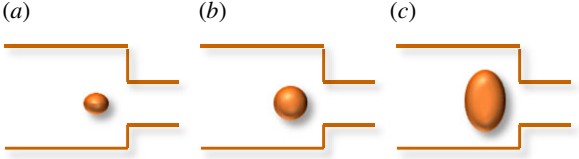

*(a)* *(b)* *(c)*

**Figure 2.** The relationship between the size of the emulsion droplets and the diameter of the pore throat.

**Table 3.** Basic core parameters under different water cut.

| pore diameter (μm) | water cut (%) | diameter of emulsion (μm) | De/Dp | outlet size of emulsion (μm) |
| --- | --- | --- | --- | --- |
| 1.804 (20 mD) | 30 | 1.136 | 0.63 | 1.129 |
| | 40 | 1.975 | 1.09 | 1.764 |
| | 50 | 2.891 | 1.60 | 1.054 |
| | 60 | 2.131 | 1.18 | 1.332 |
| | 70 | 1.342 | 0.74 | 1.020 |
| 3.126 (40 mD) | 30 | 1.136 | 0.36 | 1.131 |
| | 40 | 1.975 | 0.63 | 1.753 |
| | 50 | 2.891 | 0.92 | 2.656 |
| | 60 | 2.131 | 0.68 | 2.032 |
| | 70 | 1.342 | 0.43 | 1.232 |
| 4.615 (60 mD) | 30 | 1.136 | 0.25 | 1.116 |
| | 40 | 1.975 | 0.43 | 1.664 |
| | 50 | 2.891 | 0.63 | 2.447 |
| | 60 | 2.131 | 0.46 | 2.003 |
| | 70 | 1.342 | 0.29 | 1.035 |
| 5.923 (80 mD) | 30 | 1.136 | 0.19 | 1.129 |
| | 40 | 1.975 | 0.33 | 1.578 |
| | 50 | 2.891 | 0.49 | 2.479 |
| | 60 | 2.131 | 0.36 | 1.912 |
| | 70 | 1.342 | 0.23 | 1.165 |

## 3.3. Change of migration resistance

### 3.3.1. Viscosity curve

The curve of viscosity of emulsion systems versus different water cut and surfactant concentrations (measured by DV2T viscometer, American Brookfield Limited Company) is shown in figure 3.

It can be seen from figure 3 that when water cut is below 50%, the viscosity of the emulsion rises with the increase in water cut. The viscosity of the emulsion reaches the maximum value when the water cut reaches 50%. After that, the viscosity decreases while the water cut continues to rise. Therefore, it is considered that the water cut 50% is the transition point of the emulsion. The viscosity of the emulsions increases with the increases in the surfactant concentration.

### 3.3.2. Resistance coefficient curve

The curves of resistance coefficient $F_R$ for different emulsion systems with different water cut and surfactant concentrations flowing through cores with different permeability are shown in figures 4–11. It can be seen from the diagrams that the emulsions flowing through cores with different permeability in different water cut and surfactant concentrations show similar migration characteristics. The maximum migration resistance is higher because of similar diameter of the emulsion droplet and the pore throat when De/Dp ≈ 1. The peak value is far greater than the resistance of other emulsions,

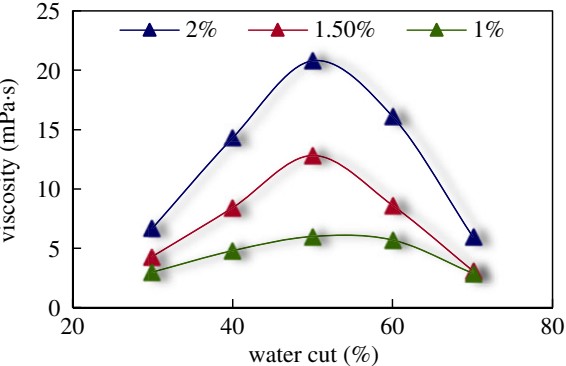

**Figure 3.** Viscosity of emulsions under different surfactant concentrations.

**Table 4.** Basic core parameters under different surfactant concentrations.

| pore diameter ($\mu$m) | surfactant concentration (%) | diameter of emulsion ($\mu$m) | De/Dp | outlet size of emulsion ($\mu$m) |
|---|---|---|---|---|
| 1.804 (20 mD) | 1 | 2.891 | 1.60 | 1.129 |
| | 1.5 | 2.437 | 1.35 | 1.764 |
| | 2 | 1.963 | 1.09 | 1.054 |
| 3.126 (40 mD) | 1 | 2.891 | 0.93 | 2.621 |
| | 1.5 | 2.437 | 0.78 | 2.142 |
| | 2 | 1.963 | 0.63 | 1.752 |
| 4.615 (60 mD) | 1 | 2.891 | 0.63 | 2.634 |
| | 1.5 | 2.437 | 0.53 | 2.042 |
| | 2 | 1.963 | 0.43 | 1.664 |
| 5.923 (80 mD) | 1 | 2.891 | 0.49 | 2.631 |
| | 1.5 | 2.437 | 0.41 | 1.942 |
| | 2 | 1.963 | 0.33 | 1.578 |

showing a larger and earlier fluctuation. The reason is that the plugging of emulsion and the separation of oil and water need to exceed the critical pressure of the liquid drop. With more and more emulsion injected, the droplets gather and block the pores, resulting in additional pressure gradient and higher displacement pressure. When the pressure increases to a certain value, the emulsion passes through the pores after deformation. The accumulation and plugging of the emulsion droplets can occur again, and the seepage resistance increases.

The resistance of the emulsion is great when De/Dp > 1, which is because the larger the emulsion droplet, the stronger the Jamin effect of the porous medium, and the higher the shear strength of the large liquid droplets. The emulsion droplets are difficult to pass through the pores. With increasing injection volume, the droplets gather in the pores and the seepage resistance increases rapidly. When the critical pressure for deformation is reached, the emulsion suffers serious dispersion and demulsification, and passes through the pores by turning into smaller droplets or causing separation of oil and water.

The diameter of the emulsion droplet is small and easy to transport in the porous medium when De/Dp < 1. Even small force is enough to push the droplet through the pore throat and wash it out of the porous medium by the post-flushing fluid. In this case, the tendency of blocking is reduced and the emulsion can move freely in pores. The larger the pore is, the less the migration resistance is. A large number of droplets enter into the porous medium and are compressed. The migration resistance of the emulsion increases initially at a certain speed. With increasing injected volume, the migration resistance increased at first and then stabilized with slight fluctuation. However, due to the adsorption

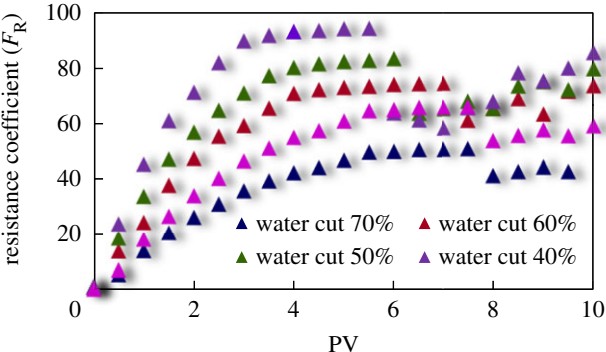

**Figure 4.** $F_R$ when emulsions with different water cut flow through 20 mD core.

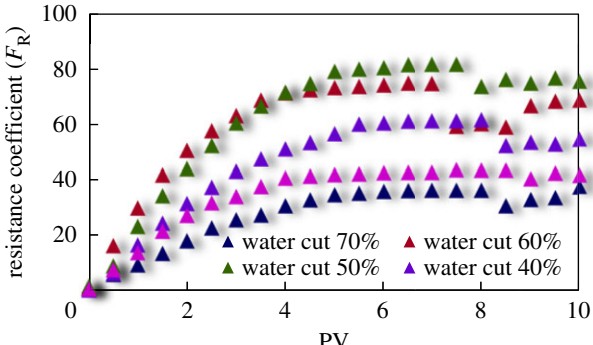

**Figure 5.** $F_R$ when emulsions with different water cut flow through 40 mD core.

on the pore surface, increasing injection of emulsion will also lead to an increase in emulsion migration resistance. However, the peak value is relatively low with no obvious fluctuation.

Generally, small pore throat is blocked at first when the emulsion transports in the porous medium. As more and more small pore throats are blocked, the flow resistance of the follow-up emulsion increases. For similar pore throat condition, the blocking capacity of small droplets is less than that of large droplets.

## 3.4. Classification of emulsion system

The emulsions are classified according to the resistance coefficient $F_R$: when $F_R$ is 80–100, the strength grade of emulsion is 3 (strong emulsion) and is expressed in △; when $F_R$ is 60–80, the strength grade of emulsion is 2 (middle emulsion) and is expressed in □; when $F_R$ is 20–60, the strength grade of emulsion is 1 (weak emulsion) and is expressed in ○. The corresponding emulsion systems with different water cut and permeability are listed in table 5.

It can be seen from table 5 that when the permeability and saturation are different, the emulsion formed in the formation is different from the injected emulsion. Firstly, the influence of the resistance coefficient means the particle size matching relationship on the classification of the emulsion. There is the biggest $F_R$ and the highest level when the ratio of the diameter of the droplet to the pore diameter is about 1. Secondly, the effect of viscosity on the classification of emulsion is considered. When the permeability is fixed, the influence of water cut on the grade of emulsion is related to the viscosity of the chemical agent injected. The greater the viscosity is, the higher the level of the emulsion is. Finally, considering the effect of permeability, the higher the permeability is, the lower the emulsion is when the water cut is fixed.

## 3.5. Prediction of the adjustment of the interlayer contradiction

According to table 5, it can be predicted which level of emulsion will be formed in the layer and whether profile control could occur in the layer at different permeability, with different injected chemical agents,

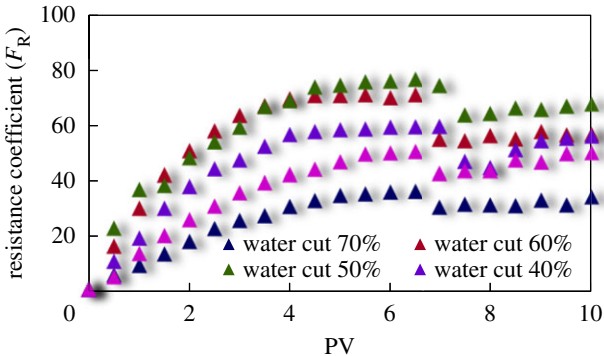

**Figure 6.** $F_R$ when emulsions with different water cut flow through 60 mD core.

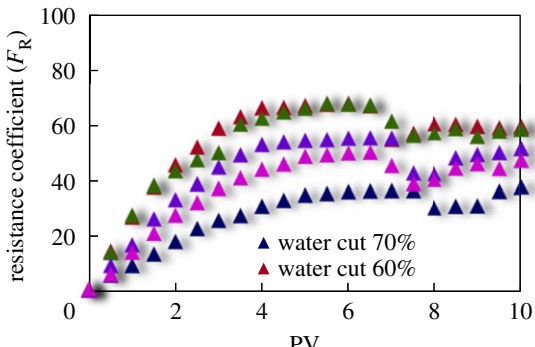

**Figure 7.** $F_R$ when emulsions with different water cut flow through 80 mD core.

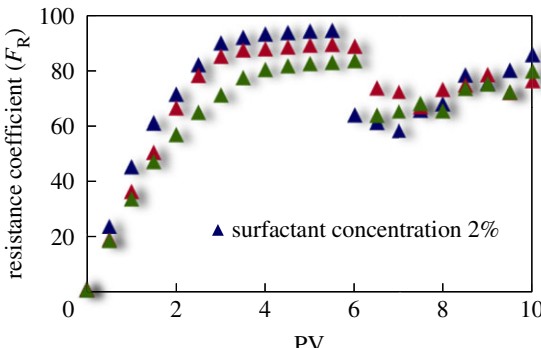

**Figure 8.** $F_R$ when emulsions with different surfactant concentrations flow through 20 mD core.

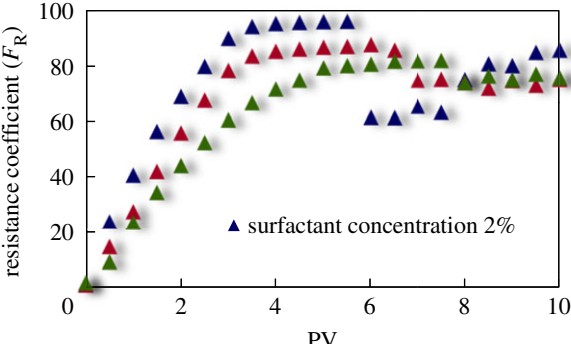

**Figure 9.** $F_R$ when emulsions with different surfactant concentrations flow through 40 mD core.

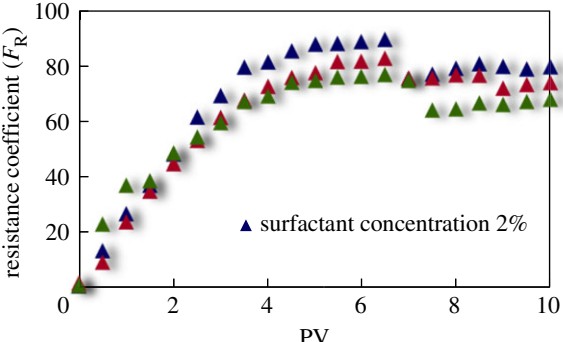

**Figure 10.** $F_R$ when emulsions with different surfactant concentrations flow through 60 mD core.

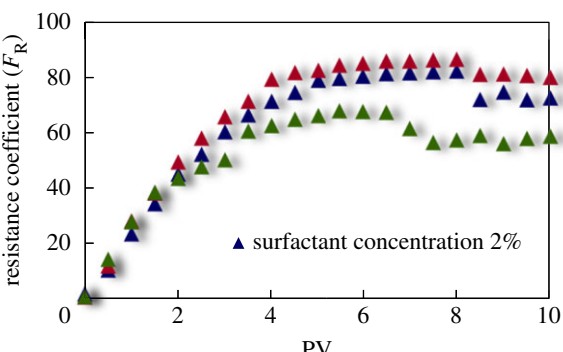

**Figure 11.** $F_R$ when emulsions with different surfactant concentrations flow through 80 mD core.

**Table 5.** Emulsion grading plate.

| permeability (mD) | surfactant concentration (%) | water cut (%) | | | | |
|---|---|---|---|---|---|---|
| | | 30 | 40 | 50 | 60 | 70 |
| 20 | 1 | □ | △ | △ | □ | ○ |
| | 1.5 | □ | △ | △ | □ | ○ |
| | 2 | △ | △ | △ | △ | □ |
| 40 | 1 | ○ | □ | △ | □ | ○ |
| | 1.5 | □ | □ | △ | □ | ○ |
| | 2 | △ | △ | △ | △ | □ |
| 60 | 1 | ○ | ○ | □ | □ | ○ |
| | 1.5 | □ | □ | △ | □ | ○ |
| | 2 | □ | △ | △ | △ | □ |
| 80 | 1 | ○ | ○ | □ | □ | ○ |
| | 1.5 | ○ | □ | △ | □ | ○ |
| | 2 | □ | □ | △ | □ | ○ |

and different water saturation of formation. Figures 12 and 13 illustrate the contribution of emulsion grading to improve the heterogeneity of the reservoirs.

Figure 12 is a schematic diagram of the contribution of emulsion grading to profile control for permeability contrast of 1.5. The formation is simulated with two parallel cores of 40 and 60 mD, with 40 and 50% of water saturation for the low and high permeability core, respectively. Firstly, weak emulsifying agent was injected into the parallel core (1%). It can be deduced that the emulsion forming in the low and high permeability cores are No. 2 grade. Therefore, the injected oil

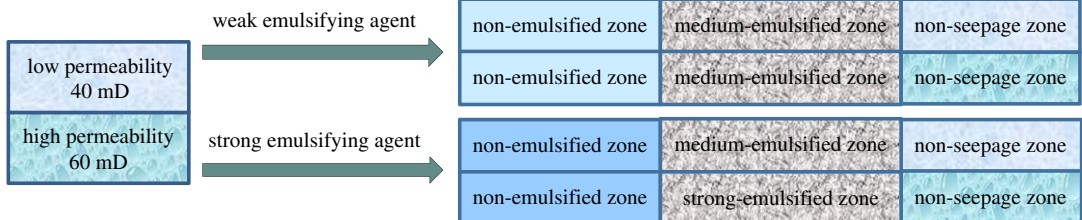

**Figure 12.** Profile control efficacy when the permeability contrast is 1.5.

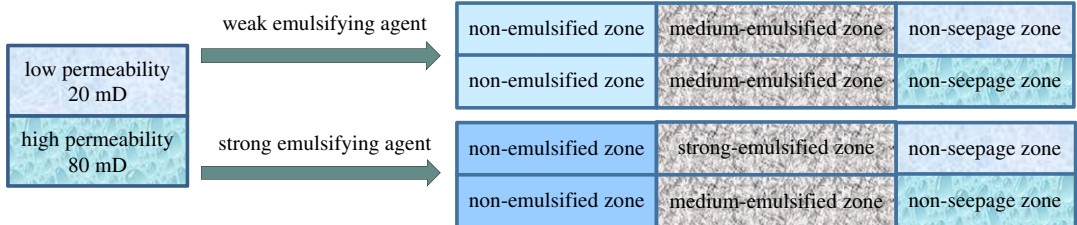

**Figure 13.** Profile control efficacy when the permeability contrast is 4.

displacement agent does not work for profile control. Then strong emulsifying agent into the parallel core (1.5%) was injected. It is shown that the emulsion forming in the low permeability core is No. 2 grade and the high permeability core is No. 3 grade. The emulsion of No. 3 grade could block the high permeability layer and force the liquid transporting and expanding the sweeping volume. The injected oil displacement agent plays a positive role in the profile control.

Similarly, figure 13 is a schematic diagram of the contribution of emulsion grading to profile control for permeability contrast of 4. The formation is simulated with two parallel cores of 20 and 80 mD, and 30% and 60% of water saturation for the low and high permeability core, respectively. Firstly, weak emulsifying agent into the parallel core (1%) is injected. It can be deduced that the emulsion forming in the low and high permeability cores are No. 2 grade. Therefore, the injected oil displacement agent does not help for the profile control of the formation. Then strong emulsifying agent into the parallel core (2%) is injected. It is shown that the emulsion forming in the low permeability core is No. 3 grade and the high permeability core is No. 2 grade. The emulsion of No. 3 grade will block the low permeability layer which is useless and the sweeping volume will be reduced. The injected oil displacement agent plays a negative role in profile control.

Overall, emulsification has a positive effect on the enhancement of recovery when the permeability contrast is small. With increasing emulsion grade (the concentration of chemical agent), the efficiency of recovery is improved. However, the emulsification has a negative effect on the enhancement of recovery when the permeability contrast is larger. With increasing emulsion grade, the negative effect becomes more obvious. It shows that when the pressure rises to a certain level in heterogeneous reservoir, the layer with low permeability starts to produce oil. The viscosifying action of emulsion has positive effect on the recovery with more oil drops being dragged into the water phase after emulsification. However, when viscosifying action of emulsion is too large, the mobility control will be worse, which is equivalent to further increasing heterogeneity of the oil layer. Lower water saturation in low permeability layer causes faster emulsification and greater viscosifying action. Higher water saturation in high permeability layer causes slower emulsification and weaker viscosifying action. All these would affect sweeping volume increase and oil displacement efficiency enhancement.

# 4. Conclusion

(1) The emulsion droplets migrate in the porous medium through deformation or broken into smaller droplets or demulsification. When the diameter of the emulsion droplets is similar to the pore diameter of the core, the maximum migration resistance of the emulsion is the highest in the porous medium with a strong blocking ability to the pores. The blocking ability is poor when the sizes of the two are poor.

(2) The blockage of large liquid drops is strong. After entering the porous medium, the large droplet blocks both pore throats which the small droplets could and could not block. Therefore, the

emulsion with large droplets has a strong blocking ability to the porous medium with larger migration resistance.

(3) The emulsion formed in the formation is different to the injected when the permeability and saturation are different. This is related to resistance coefficient, viscosity of chemical agent and permeability. The higher the resistance coefficient is, the higher the viscosity of the chemical agent is, the higher the permeability is and the higher the grade of the emulsion is.

(4) The emulsification has a positive contribution to the enhancement of recovery when the permeability contrast is small. With increasing emulsion grade (the concentration of chemical agent), the recovery efficiency is increased. However, the emulsification has a negative effect on the enhancement of recovery when the permeability contrast is bigger. With increasing emulsion grade, the negative effect is more obvious.

Data accessibility. The datasets supporting this article have been provided as the electronic supplementary material. All data for this work are presented in tables 1–3.

Authors' contributions. The laboratory work and the sequence alignments were carried out by T.Z. and T.D. who also participated in data analysis and the design of the study, and drafted the manuscript; Y.W. and X.L. performed the statistical analyses; C.F. and W.Z. collected field data; B.H. conceived of the study, designed the study, coordinated the study and helped drafting the manuscript. All authors gave final approval for publication.

Competing interests. We declare that we have no competing interests.

Funding. This work was financially sponsored by National Natural Science Foundation of China (no. 51804077).

Acknowledgements. The authors thank the editors and the reviewers for their very helpful comments, which led to the better presentation of the ideas proposed in this manuscript.

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
