## [Reviewer comments · Royal Society Open Science]

Review History

RSOS-181634.R0 (Original submission)

Review form: Reviewer 1

Is the manuscript scientifically sound in its present form?

No

Are the interpretations and conclusions justified by the results?

No

Is the language acceptable?

No

Is it clear how to access all supporting data?

No

Do you have any ethical concerns with this paper?

No

Have you any concerns about statistical analyses in this paper?

I do not feel qualified to assess the statistics

Recommendation?

Reject

Comments to the Author(s)

The authors tried to evaluate how to effectively divert the fluids into porous media the way they want to have. The emulsion is used for achieving such purpose. Due to following concerns, I do not feel comfortable with recommending publishing this manuscript.

1) The definition of class III reservoir is not given and this makes the purpose of the article blurry.

2) Emulsions have been proposed as a kind of diversion fluids long time ago in the industry.

After reviewing the manuscript, I am getting confused with the novelty of the work. Many concepts are well known in the textbooks on enhanced oil recovery.

3) How to upscale the results to field? No dimensionless analysis is performed, making it hard to upscale it to field scale.

4) Emulsions are normally classified according to the Winsor phase diagrams. I do not see authors have done such a job to classify the emulsions prepared by them. They simply classify them as weak, strong or medium strong emulsion, which is not quite useful in terms of engineering use. More careful experimental design and problem statement is lacking in the current manuscript.

5) Overall, the writing of the manuscript is not acceptable. There are a lot of grammar errors and confusing statements. A thorough revision should be in place to make it more readable and suitable for journal publication.

Review form: Reviewer 2**Is the manuscript scientifically sound in its present form?**

Yes

Are the interpretations and conclusions justified by the results?

Yes

Is the language acceptable?

Yes

Is it clear how to access all supporting data?

Yes

Do you have any ethical concerns with this paper?

No

Have you any concerns about statistical analyses in this paper?

No

Recommendation?

Accept with minor revision (please list in comments)

Comments to the Author(s)

This article studies the flow mechanisms and profile control of emulsion system in reservoirs. The experimental work was well designed and the results were analyzed and discussed. The interpretations of research results are thorough and rigorous, and conclusions are justified.

A few comments are as follows:

Line 22 on Page 1: "flows" --"flowing"

Line 51 on Page 4: "Middle container" -"accumulator"

Line 50 on Page 6: "movement" --"flow"?, and many other "movement" in the paper

Line 20 on Page 11: remove "molecular".

Review form: Reviewer 3

Is the manuscript scientifically sound in its present form?

Yes

Are the interpretations and conclusions justified by the results?

Yes

Is the language acceptable?

Yes

Is it clear how to access all supporting data?

Yes

Do you have any ethical concerns with this paper?

No

Have you any concerns about statistical analyses in this paper?

No

Recommendation?

Accept with minor revision (please list in comments)

Comments to the Author(s)

The authors presented a study of the effect of oil emulsification on the migration performance. They did a series of work to elucidate the matching relations between stable emulsion and oil layer and made a matching chart to predict parallel-core displacement results with different permeability contrasts. This paper encompasses a noticeable amount of experimental results, it can contribute to the literature at these conditions. I suggest the minor revision for publishing this paper.

- (1) It is better to revise the abstract to give a better description. Specifically, the sentence "Through core displacement experiment, the pressure, diversion rate, water cut and recovery factor of the parallel core in different contrasts are measured" is very confusing and in fact, in this paper, no displacement experiment is found.
- (2) Para 2 of section 2.2.3 should be simplified and the core permeability is not match with section 3.2, please check it carefully.
- (3) In section 3.5, the prediction of profile control with different permeability contrasts is confusing. Please check it carefully.
- (4) The language of this article is suggested to be improved.

Decision letter (RSOS-181634.R0)

12-Mar-2019

Dear Miss Zhu:

Title: The Efficacy of Migration and Profile Control with Emulsion Systems in Class III Reservoirs
Manuscript ID: RSOS-181634

The editor assigned to your manuscript has now received comments from reviewers. We would like you to revise your paper in accordance with the referee and Subject Editor suggestions which can be found below (not including confidential reports to the Editor). Please note this decision does not guarantee eventual acceptance. I apologise that this has taken longer than usual.

Please submit your revised paper before 04-Apr-2019. Please note that the revision deadline will expire at 00.00am on this date. If we do not hear from you within this time then it will be assumed that the paper has been withdrawn. In exceptional circumstances, extensions may be possible if agreed with the Editorial Office in advance. We do not allow multiple rounds of revision so we urge you to make every effort to fully address all of the comments at this stage. If deemed necessary by the Editors, your manuscript will be sent back to one or more of the original reviewers for assessment. If the original reviewers are not available we may invite new reviewers.

On behalf of the Subject Editor Professor Anthony Stace and the Associate Editor Professor Hazel Cox.

RSC Associate Editor:
Comments to the Author:
(There are no comments.)

RSC Subject Editor:
Comments to the Author:
(There are no comments.)

Reviewers' Comments to Author:
Reviewer: 1

Comments to the Author(s)

The authors tried to evaluate how to effectively divert the fluids into porous media the way they want to have. The emulsion is used for achieving such purpose. Due to following concerns, I do not feel comfortable with recommending publishing this manuscript.

- 1) The definition of class III reservoir is not given and this makes the purpose of the article blurry.
- 2) Emulsions have been proposed as a kind of diversion fluids long time ago in the industry. After reviewing the manuscript, I am getting confused with the novelty of the work. Many concepts are well known in the textbooks on enhanced oil recovery.
- 3) How to upscale the results to field? No dimensionless analysis is performed, making it hard to upscale it to field scale.
- 4) Emulsions are normally classified according to the Winsor phase diagrams. I do not see authors have done such a job to classify the emulsions prepared by them. They simply classify them as weak, strong or medium strong emulsion, which is not quite useful in terms of engineering use. More careful experimental design and problem statement is lacking in the current manuscript.
- 5) Overall, the writing of the manuscript is not acceptable. There are a lot of grammar errors and confusing statements. A thorough revision should be in place to make it more readable and suitable for journal publication.

Reviewer: 2

Comments to the Author(s)

This article studies the flow mechanisms and profile control of emulsion system in reservoirs. The experimental work was well designed and the results were analyzed and discussed. The interpretations of research results are thorough and rigorous, and conclusions are justified.

A few comments are as follows:

Line 22 on Page 1: "flows" --"flowing"

Line 51 on Page 4: "Middle container" --"accumulator"

Line 50 on Page 6: "movement" --"flow"?, and many other "movement" in the paper

Line 20 on Page 11: remove "molecular".

Reviewer: 3

Comments to the Author(s)

The authors presented a study of the effect of oil emulsification on the migration performance. They did a series of work to elucidate the matching relations between stable emulsion and oil layer and made a matching chart to predict parallel-core displacement results with different permeability contrasts. This paper encompasses a noticeable amount of experimental results, it can contribute to the literature at these conditions. I suggest the minor revision for publishing this paper.

(1) It is better to revise the abstract to give a better description. Specifically, the sentence "Through core displacement experiment, the pressure, diversion rate, water cut and recovery factor of the parallel core in different contrasts are measured" is very confusing and in fact, in this paper, no displacement experiment is found.

(2) Para 2 of section 2.2.3 should be simplified and the core permeability is not match with section 3.2, please check it carefully.

(3) In section 3.5, the prediction of profile control with different permeability contrasts is confusing. Please check it carefully.

(4) The language of this article is suggested to be improved.

Author's Response to Decision Letter for (RSOS-181634.R0)

See Appendices A - C.

RSOS-181634.R1 (Revision)

Review form: Reviewer 3

Is the manuscript scientifically sound in its present form?

Yes

Are the interpretations and conclusions justified by the results?

Yes

Is the language acceptable?

Yes

Is it clear how to access all supporting data?

Yes

Do you have any ethical concerns with this paper?

No

Have you any concerns about statistical analyses in this paper?

No

Recommendation?

Accept as is

Comments to the Author(s)

The author has revised it according to my review. I think it can be published in ROYAL SOCIETY OPEN SCIENCE.

Decision letter (RSOS-181634.R1)

15-Apr-2019

Dear Miss Zhu:

Title: The Efficiency of Migration and Profile Control with Emulsion Systems in Class III Reservoirs
Manuscript ID: RSOS-181634.R1

It is a pleasure to accept your manuscript in its current form for publication in Royal Society Open Science. The chemistry content of Royal Society Open Science is published in collaboration with the Royal Society of Chemistry.

Reviewer(s)' Comments to Author:
Reviewer: 3

Comments to the Author(s)

The author has revised it according to my review. I think it can be published in ROYAL SOCIETY OPEN SCIENCE.

Appendix A

Response to the Comments of Reviewer 1:

First of all, thank you very much for your comments on this article. We tried our best to improve the manuscript according to your comments and made some changes in the manuscript. And the replies to your comments are as follows. We appreciate for your warm work earnestly, and hope that the correction will meet with approval. Once again, thank you very much for your comments and suggestions.

The authors tried to evaluate how to effectively divert the fluids into porous media the way they want to have. The emulsion is used for achieving such purpose. Due to following concerns, I do not feel comfortable with recommending publishing this manuscript.

Point 1: The definition of class III reservoir is not given and this makes the purpose of the article blurry.

Response 1: Special thanks to the problems which you proposed. According to your suggestion, we have given the definition of class III reservoir in the abstract: reservoir with permeability less than 100 mD, so as to make the purpose of the article clearer. The emulsification in formation is often weak and unstable, the emulsification of displacement agent is intermittent, and the regularity is not obvious enough, which brings some difficulties to the research. Especially in the low permeability formation (class III reservoir), the study of emulsification is more difficult. Therefore, we prepared a stable emulsion made by surfactant and kerosene to magnify the emulsification and studied its migration process in class III reservoir, then predicted the emulsification effect of displacement agent in formation, and analyzed the profile control effect of class III reservoir. We hope you will be satisfied with our reply and reconsider our article.

Point 2: Emulsions have been proposed as a kind of diversion fluids long time ago in the industry. After reviewing the manuscript, I am getting confused with the novelty of the work. Many concepts are well known in the textbooks on enhanced oil recovery.

Response 2: It is really true as you proposed that It is really true as you proposed that emulsions have been proposed as a kind of diversion fluids long time ago, and many concepts are well known about enhanced oil recovery. But our research still has the following three innovations.

First of all, we prepared a stable emulsion made by surfactant and kerosene to magnify the emulsification and studied its migration process in class III reservoir, then predicted the emulsification effect of displacement agent in formation, and analyzed the profile control effect of class III reservoir. The emulsion is milky white and stable, it did not occur oil-water separation within 72 hours, and oil droplet coalescence was not found under microscopic observation, which is very helpful to study the migration

resistance of the emulsion in the core.

Secondly, we defined two concepts, one is the ratio (the ratio of emulsion droplet diameter to core pore throat diameter), the other is the resistance coefficient (the ratio of emulsion to water migration pressure in core). It is found that the closer the ratio is to 1, the larger the resistance coefficient is, which indicates that the matching relationship between emulsion droplet diameter and core pore throat diameter is great.

Finally, we classify the stable emulsion into three grades according to the resistance coefficient of the emulsion on account of the commonly used classification criteria of an oil production plant in Daqing: weak emulsion, medium emulsion and strong emulsion. The grading plant of surfactant concentration, water content and emulsification level was made, the emulsification level when injecting surfactant with different concentration into different permeability formation and driving the formation to different water content was predicted, and the profile control effect of emulsification on heterogeneous formation was analyzed. We hope you will be satisfied with our reply and reconsider our article.

Point 3: How to upscale the results to field? No dimensionless analysis is performed, making it hard to upscale it to field scale.

Response 3: Special thanks to the problems which you proposed. The reference value of this paper for the field is not the dosage of surfactant, but the prediction of emulsification caused by displacement agent in formation, and the analysis of profile control of class III reservoir.

Table.1 Emulsion grading plate

Permeability (mD)	Surfactant concentration (%)	Water cut (%)				
		30	40	50	60	70
20	1	■	△	△	□	○
	1.5	▲	△	△	□	○
	2	△	△	△	△	□
40	1	○	■	△	□	○
	1.5	□	■	△	□	○
	2	△	△	△	△	□
60	1	○	○	■	□	○
	1.5	□	□	▲	□	○
	2	□	△	△	△	□
80	1	○	○	□	■	○
	1.5	○	□	△	□	○
	2	□	□	△	■	○

According to the Table.1 and as shown in Fig. 1, when the permeability contrast is small (permeability is 40 mD and 60 mD, respectively), a weak emulsifying displacement agent (surfactant concentration is 1%) is injected into the parallel cores, and the water content of the core are 40% and 50%, respectively. It can be inferred that

the medium emulsion (green background, blue mark) will be produced in the low permeability and high permeability cores. Therefore, the injected displacement agent does not play a profile control role in the formation.

When a strong emulsifying displacement agent (surfactant concentration is 1.5%) is injected into the parallel cores, and the water content of the core are 40% and 50%, respectively. It can be inferred that the medium emulsion (green background, blue mark) will be produced in the low permeability core and the strong emulsion (pink background, blue mark) will be produced in the high permeability core. The strong emulsion will block the high permeability layer, promote the fluid flow diversion and expand the sweep volume, so the injected displacement agent plays an active role in profile control of the formation. Therefore, it is suggested to inject strong emulsifying displacement agent into the formation with smaller permeability contrast.

Fig.1 Profile control efficacy when the permeability contrast is 1.5

As shown in Fig. 2, when the permeability contrast is bigger (permeability is 20 mD and 80 mD, respectively), a weak emulsifying displacement agent (surfactant concentration is 1%) is injected into the parallel cores, and the water content of the core are 30% and 60%, respectively. It can be inferred that the medium emulsion (green background, blue mark) will be produced in the low permeability and high permeability cores. Therefore, the injected displacement agent does not play a profile control role in the formation.

When a strong emulsifying displacement agent (surfactant concentration is 2%) is injected into the parallel cores, and the water content of the core are 30% and 60%, respectively. It can be inferred that the strong emulsion (pink background, red mark) will be produced in the low permeability core and the medium emulsion (green background, red mark) will be produced in the high permeability core.

The strong emulsion will block the low permeability layer, make the low permeability layer unusable and diminish the sweep volume, so the injected displacement agent plays a negative role in profile control of the formation. Therefore, it is suggested to inject weak emulsifying displacement agent into the formation with bigger permeability contrast.

Fig.2 Profile control efficacy when the permeability contrast is 4

The prediction results can provide some guidance for the selection of emulsifying strength of displacement agents in class III reservoir. We hope you will be satisfied with our reply and reconsider our article.

Point 4: Emulsions are normally classified according to the Winsor phase diagrams. I do not see authors have done such a job to classify the emulsions prepared by them. They simply classify them as weak, strong or medium strong emulsion, which is not quite useful in terms of engineering use. More careful experimental design and problem statement is lacking in the current manuscript.

Response 4: Special thanks to the problems which you proposed. First of all, according to your question, we consulted the relevant literature and found that the Winsor phase diagrams are commonly used as the classification standard of microemulsion. But the solution we used is a stable emulsion prepared by surfactant and kerosene, without adding salt and alcohol, which is different from microemulsion. Secondly, as you said, the classification of emulsions in the plate is simple, but we classify the emulsions according to the commonly used classification standards of an oil production plant in Daqing, and the original intention of the plate is to show the relationship between the ratio and the resistance coefficient more clearly, and indicate the prediction function of the plate. We hope you will be satisfied with our reply and reconsider our article.

Point 5: Overall, the writing of the manuscript is not acceptable. There are a lot of grammar errors and confusing statements. A thorough revision should be in place to make it more readable and suitable for journal publication.

Response 5: Special thanks to the problems which you proposed. We have asked our American friend to revise the English writing. We have revised the introduction and abstract of the paper. And here we did not list the changes but marked in revised paper. We hope you will be satisfied with our reply and reconsider our article.

Once again, thank you very much for your comments and suggestions.

Appendix B

Response to the Comments of Reviewer 2:

First of all, thank you very much for your comments on this article. We tried our best to improve the manuscript according to your comments and made some changes in the manuscript. And the replies to your comments are as follows. We appreciate for your warm work earnestly, and hope that the correction will meet with approval. Once again, thank you very much for your comments and suggestions.

This article studies the flow mechanisms and profile control of emulsion system in reservoirs. The experimental work was well designed and the results were analyzed and discussed. The interpretations of research results are thorough and rigorous, and conclusions are justified. A few comments are as follows:

Point 1: Line 22 on Page 1: “flows” → “flowing”.

Response 1: Special thanks to the problems which you proposed. We have corrected the error according to your comments in paper.

Point 2: Line 51 on Page 4: “Middle container” → “accumulator”.

Response 2: Special thanks to the problems which you proposed. We have corrected the error according to your comments in paper.

Point 3: Line 50 on Page 6: “movement” → “flow” ? and many other “movement” in the paper.

Response 3: Special thanks to the problems which you proposed. We have corrected all the “movement” to “migration” in paper, is it appropriate? I am pleased to listen your suggestion.

Point 4: Line 20 on Page 11: remove “molecular”.

Response 4: Special thanks to the problems which you proposed. We have corrected the error according to your comments in paper.

Once again, thank you very much for your comments and suggestions.

Appendix C

Response to the Comments of Reviewer 3:

First of all, thank you very much for your comments on this article. We tried our best to improve the manuscript according to your comments and made some changes in the manuscript. And the replies to your comments are as follows. We appreciate for your warm work earnestly, and hope that the correction will meet with approval. Once again, thank you very much for your comments and suggestions.

The authors presented a study of the effect of oil emulsification on the migration performance. They did a series of work to elucidate the matching relations between stable emulsion and oil layer and made a matching chart to predict parallel-core displacement results with different permeability contrasts. This paper encompasses a noticeable amount of experimental results, it can contribute to the literature at these conditions. I suggest the minor revision for publishing this paper.

Point 1: It is better to revise the abstract to give a better description. Specifically, the sentence “Through core displacement experiment, the pressure, diversion rate, water cut and recovery factor of the parallel core in different contrasts are measured” is very confusing and in fact, in this paper, no displacement experiment is found.

Response 1: Special thanks to the problems which you proposed. We have revised the introduction and abstract of the paper. And here we did not list the changes but marked in revised paper.

Point 2: Para 2 of section 2.2.3 should be simplified and the core permeability is not match with section 3.2, please check it carefully.

Response 2: Special thanks to the problems which you proposed. We have simplified section 2.2.3 and corrected the core permeability according to your comments in paper.

Point 3: In section 3.5, the prediction of profile control with different permeability contrasts is confusing. Please check it carefully.

Response 3: Special thanks to the problems which you proposed. We have corrected the error according to your comments in paper.

Point 4: The language of this article is suggested to be improved.

Response 4: Special thanks to the problems which you proposed. We have asked our American friend to revise the English writing. And here we did not list the changes but marked in revised paper.

Once again, thank you very much for your comments and suggestions.